# Linear Versus Nonlinear Acoustic Probing of Plasticity in Metals: A Quantitative Assessment

**DOI:** 10.3390/ma11112217

**Published:** 2018-11-08

**Authors:** Carolina Espinoza, Daniel Feliú, Claudio Aguilar, Rodrigo Espinoza-González, Fernando Lund, Vicente Salinas, Nicolás Mujica

**Affiliations:** 1Departamento de Física, Facultad de Ciencias Físicas y Matemáticas, Universidad de Chile, Avenida Blanco Encalada 2008, Santiago 8370449, Chile; dafeliu@ug.uchile.cl (D.F.); flund@dfi.uchile.cl (F.L.); vicente.salinas@ing.uchile.cl (V.S.); nmujica@dfi.uchile.cl (N.M.); 2Departamento de Ingeniería Metalúrgica y Materiales, Universidad Técnica Federico Santa Maria, Av. España 1680, Valparaíso 2390123, Chile; claudio.aguilar@usm.cl; 3Departamento de Ingeniería Química, Biotecnología y Materiales, Facultad de Ciencias Físicas y Matemáticas, Universidad de Chile, Avenida Beauchef 851, Santiago 8370456, Chile; roespino@ing.uchile.cl; 4Núcleo de Matemáticas, Física y Estadística, Facultad de Estudios Interdisciplinarios, Universidad Mayor, Manuel Montt 318, Providencia 7500628, Chile

**Keywords:** alloys, nondestructive testing, dislocation density, plasticity, ultrasound, nonlinear acoustics

## Abstract

The relative dislocation density of aluminum and copper samples is quantitatively measured using linear Resonant Ultrasound Spectroscopy (RUS). For each metallic group, four samples were prepared with different thermomechanical treatments in order to induce changes in their dislocation densities. The RUS results are compared with Nonlinear Resonant Ultrasound Spectroscopy (NRUS) as well as Second Harmonic Generation (SHG) measurements. NRUS has a higher sensitivity by a factor of two to six and SHG by 14–62%. The latter technique is, however, faster and simpler. As a main result, we obtain a quantitative relation between the changes in the nonlinear parameters and the dislocation density variations, which in a first approximation is a linear relation between these differences. We also present a simple theoretical expression that explains the better sensitivity to dislocation content of the nonlinear parameters with respect to the linear ones. X-Ray diffraction measurements, although intrusive and less accurate, support the acoustics results.

## 1. Introduction

Plastic behavior of metallic materials is determined by dislocations, with the transition from brittle to ductile behavior being of particular interest. Dislocation density is then a key variable in order to assess the deformation state of a given sample or piece in service. Recently, several techniques have enabled the in situ study of plastic behavior and, directly or indirectly, these techniques enable the quantification of dislocation density. This has been done at the micro and nano scale, using transmission electron microscopy (TEM), scanning electron microscopy (SEM) and atomic force microscopy (AFM). For example, Oh et al. [1] have reported in situ observations of dislocation nucleation and escape; Landau et al. [2] have studied dislocation patterning; Zhang et al. [3] have reported real-time correlation between flow stress and dislocation density; and Du et al. [4] have reported observations of dislocation emission. However, these are destructive techniques and, in general, small, specially prepared samples are required. In most engineering applications these conditions can not be satisfied. Therefore, in situ and non-destructive tests are desirable.

Acoustics has long been a tool for the non-destructive evaluation of materials [5,6,7,8]. It is routinely used for crack detection [9,10,11,12]. However, concerning the plastic behavior of metals and alloys, it is only recently that progress in theoretical modeling and instrumentation development have enabled acoustic measurements to emerge as a quantitative tool to measure dislocation density. On the theory side, Maurel et al. [13,14], building on the classic work of Granato and Lücke [15] derived the following formula, valid for isotropic materials, that relates the change in dislocation density between two samples with the change in the speed of shear waves:(1)$$\frac{\Delta v_{T}}{v_{T}}=-\frac{8}{5\pi^{4}}\Delta \left(nL^{3}\right)=-\frac{8}{5\pi^{4}}\Delta \left(\Lambda L^{2}\right),$$
where $\Delta v_{T}/v_{T}$ is the relative change of shear wave velocity between two samples of a material that differ in dislocation density Λ=nL, and *n* is the number of dislocation segments of (average) length *L* per unit volume. This is an extremely simple result that was experimentally verified using Resonant Ultrasound Spectroscopy (RUS) [16] by Mujica et al. [17]. In addition, Salinas et al. [18] measured $nL^{3}$ in-situ, and continuously, as a function of applied stress for aluminum under standard testing conditions. These measurements provided an experimental verification of Taylor’s rule, which relates flow stress with dislocation density, with unprecedented accuracy [18]. They provide a solid basis to use velocity measurements as a nonintrusive quantitative measure, as opposed to qualitative estimate, changes in dislocation density. From a purely conceptual point of view, it is interesting to notice that the relevant dimensionless parameter that measure dislocation density is $nL^{3}$.

Armed with this new tool, we can use it to assess the accuracy of other proposed techniques to determine dislocation density. For example, nonlinear methods have been proposed because of their potentially superior sensitivity [9]. Nonlinear acoustics has been widely used to probe material properties in many different fields, such as the nondestructive testing of single crystals and homogeneus small samples [19], geomaterials [11,20,21], biomaterials [10,12,22], and thin films [23]. Nonlinear behavior has been monitored using Rayleigh waves as well [24]. There appears to be a wide agreement in the literature that nonlinear methods are quite sensitive to small-scale inhomogeneities. Can nonlinear acoustics be used to monitor dislocation proliferation in metals and alloys?

## 2. Materials and Methods

The present article provides a quantitative assesment of linear versus nonlinear acoustic measurement of dislocation density in commercially pure copper and aluminum.

One nonlinear acoustic experimental method that is widely used as a non destructive evaluation tool is Second Harmonic Generation (SHG) [25]. In this method, a second harmonic wave is generated from a propagating monochromatic elastic wave, due to the anharmonicity of the elastic material and the presence of microstructural features such as dislocations. The second harmonic nonlinear response is quantified by
(2)$$\beta =\frac{8}{xk^{2}}\frac{A_{2\omega }}{A_{\omega }^{2}},$$
where *k* is the wave number, *x* is the elastic wave propagation distance, and $A_{\omega }$ and $A_{2\omega }$ are the absolute physical displacements of the fundamental and second harmonic waves [25].

A recent review [26] reports measurements of the amplitude of the second harmonic relative to the amplitude of the first harmonic, in samples of aluminum alloy and of steel before and after plastic elongation (0.2% in aluminum, 1.5% in low carbon steel). There is an unmistakable difference, at least in part attributable to the presumed difference in dislocation density. However, there does not appear to be an accepted model that quantitatively relates this difference to a specific increase in dislocation density (see [26] and references therein).

As reported above, RUS relies on linear theory. It provides a complete set of elastic constants using one single measurement of the resonant spectrum in a given ultrasonic frequency range [27]. Extending the drive amplitude beyond the linear limit into the nonlinear regime one obtains Nonlinear Resonant Ultrasound Spectroscopy (NRUS), which is based on changes of one particular resonant frequency [9,10,11,12,28]. The corresponding frequency shift $\Delta f=f_{i}-f_{0}$ is related phenomenologically with the average strain amplitude Δϵ by the nonlinear parameter α, defined through
(3)$$\frac{\Delta f}{f_{0}}=\alpha \Delta \epsilon =\alpha \gamma V_{rec}=\alpha^{\prime }V_{rec},$$
where $f_{0}$ is the resonant frequency in the linear regime. Here, we follow Payan [9]; instead of measuring the strain ϵ, we measure the pressure sensor voltage amplitude, $V_{rec}$, so we measure the nonlinear parameter $\alpha^{\prime }$, which will differ for samples with different dislocation densities. Also, we use the method of Johnson to account for the effect of temperature [29].

In this work, two groups of aluminum and copper samples have been used to perform RUS, NRUS, SHG, as well as X-ray diffraction (XRD), measurements, the latter as a control method. We show that the results using different acoustic methods are well correlated with those obtained by XRD peak broadening profile analysis. The relative sensitivity of RUS, NRUS and SHG are presented and we show that nonlinear parameters are more sensitive to the presence of dislocations than the linear ones.

99.999 at% pure aluminum and 99.95 at% pure copper samples were used to perform RUS, NRUS, SHG and XRD measurements. Sample characteristics are given in Table 1, including parallelepiped dimensions, mass density and the thermo-mechanical treatment details. From the same as-received bar, four pieces were taken to prepare the experimental samples: all samples were cold-rolled at 82.8% and 88.3% in the aluminum and copper groups, respectively. Then, three samples of each group were annealed at 450${}^{\circ }$ for Al and 850${}^{\circ }$ for Cu, for 15, 30 and 60 min, labeled as Roll A15, Roll A30 and Roll A60 respectively. Their melting points are 659 ${}^{\circ }$C and 1083 ${}^{\circ }$C, respectively. The sample without annealing was labeled only as Roll. It is well known that annealing leads to lower dislocation density, and stronger cold-rolling leads to higher dislocation density [17]. For each one of the four pieces per group, one portion was set aside for ultrasonic testing, and another two for XRD. A schematic illustration of these samples is presented in Figure 1.

The dislocation density variations are obtained from the transverse wave speed $v_{T}$ measurements, which is done with RUS. For a correct application of this technique, the pieces must be modelable as perfect parallelepipeds to avoid resonance shifts [30]. The Al samples we analyzed had average dimensions (0.500±0.003)×(1.704±0.003)×(5.005±0.003) cm${}^{3}$, and average density 2.667±0.005 g/cm${}^{3}$. The Cu samples had average dimensions (0.399±0.005)×(1.701±0.001)×(5.001±0.003) cm${}^{3}$, and average density 8.891±0.010 g/cm${}^{3}$. Of course, precise measurements were made for each single sample in order to correctly apply the characterization methods. The XRD samples had average dimensions $\left(1.704\pm 0.003\right)\times {\left(0.500\pm 0.003\right)}^{2}$ cm${}^{3}$ and $\left(1.701\pm 0.001\right)\times {\left(0.399\pm 0.005\right)}^{2}$ cm${}^{3}$ for the Al and Cu groups, respectively.

Both the linear and the nonlinear resonant ultrasound spectroscopy used the same setup [17,31]. A schematic representation of this experimental setup, including the instrument’s brand and model, is shown in Figure 2. The positioning of the sample and its assembly conditions are the same as those described in [17]. RUS is used to measure the shear wave velocity, because the shear modulus $C_{44}$ can be determined with much higher accuracy. The drive amplitude is 1 V in the linear regime. The frequency sweep is performed between 26 kHz and 175 kHz, with 26 identified modes on average for Al. For Cu samples, the range of frequencies is 19 kHz to 127 kHz, with 21 identified modes on average. Ten independent measurements per sample were made to obtain associated statistical errors.

Imposing transverse isotropy, we have computed the anisotropy parameter $\epsilon =1-2C_{44}/\left(C_{11}-C_{12}\right)$ for both groups [32]; within experimental errors its is zero or very small for all samples. Additionally, we have computed the transverse wave speed imposing both isotropy and transverse isotropy in the RUS analysis and the differences obtained are ≲0.3%. XRD patterns show some degree of texture, which we have quantified using the March–Dollase model [33]. The March–Dollase parameters for most reflections are close to 1. However, Cu peaks (220) and Al peaks (200) have parameters smaller than 1 but with small weight factors. We finally conclude that Al and Cu samples have a low degree of texture.

For NRUS application, the set up is exactly the same as for RUS. For both the Al and Cu groups, the resonance frequency that was chosen is close to 49 kHz and 39 kHz, respectively. The exact value depends on the specific dimensions of each sample. The reason for this choice was that the selected modes were the most energetic in the frequency range studied. In the non-linear regime, we verified that the resonance is asymmetric and that its amplitude ceases to be a linear function of the excitation voltage.

The third acoustic method used in this work is SHG. In this case, the experimental setup for non-linear ultrasonic measurements is simpler than for RUS and NRUS. It is shown in Figure 3. A continuous sine wave of frequency f=3 MHz is transmitted into the material. Thus, a longitudinal wave is propagated across the length d≈1.7 cm of each sample of both groups and the resulting response is analyzed for its nonlinear features. Two equal transducers are placed on each side of the specimen (Panametrics—V110, resonant at 5 MHz, with element diameter 8.8 mm). Through Fourier analysis of the received signal, we measure the fundamental ($A_{\omega }^{\prime }$) and the second harmonic ($A_{2\omega }^{\prime }$) amplitudes, in volts.

In general, the non-linear parameter is presented in units of 1/Volts [25]. This is because precise transducer calibrations are difficult at such low driving amplitudes, which occur even in the non-linear regime. Thus, following Matlack [25], instead of calculating β in dimensionless form we measure
(4)$$\beta^{\prime }=A_{2\omega }^{\prime }/{\left(A_{\omega }^{\prime }\right)}^{2},$$
which is based on the amplitudes measured in voltage units.

XRD measurements were carried out with the same procedure and equipment reported by Salinas et al. [18] Microstructural parameters such as lattice parameter *a* and microstrain ${\langle \epsilon^{2}\rangle }^{1/2}$, were obtained from Rietveld refinements of the X-ray patterns with the Materials Analysis Using Diffraction (MAUD) software and $LaB_{6}$ (a=4.1565915(1) Å) as external standard for the determination of instrumental broadening. Using the information provided by MAUD, it is possible to obtain a measurement of dislocation density $\Lambda_{XRD}$ for each Al and Cu sample through
(5)$$\Lambda_{XRD}=\frac{24\pi E}{GF}\frac{\langle \epsilon^{2}\rangle }{a^{2}},$$
where F≈5 for FCC materials, *E* is Young’s modulus and *G* is the shear modulus. The *E* and *G* values used for Al were 74.4±1.9 GPa and 28.1±0.8 GPa, respectively. These values were calculated as an average of those reported in [27,34,35]. For Cu, *E* and *G* used were 124.5±0.7 GPa and 45.4±1.2 GPa respectively, obtained from [36]. We measured two pieces for the same sample of both groups, in order to have an associated statistical error, beyond that provided by the refinement.

## 3. Results

Figure 4 shows an example of an XRD pattern for an aluminum and copper sample. As in recent works [18], there is not one crystallite size, but rather a distribution of sizes that contribute to each diffraction peak, each one having an associated microstrain. Using the information about the volume fraction of each phase provided by MAUD, the results for Λ were calculated as a weighted average of results for differents crystallite sizes.

The results of the acoustics measurements are given in Table 2, where the behaviors of the linear and nonlinear parameters are compared and contrasted. The linear parameter $v_{T}$ shows variations between purely rolled and annealed pieces between 1.7% and 2.6% for Al, and 2.9% and 4.4% for Cu. The non-linear parameters are decreasing functions of the shear velocity $v_{T}$. This means they are increasing functions of dislocation density. The parameter $\alpha^{\prime }$ shows remarkable changes: 39% to 125% for Al, and 320% to 510% for Cu. Finally, $\beta^{\prime }$ has variations from 14% to 20% for Al, and 19% for 62% for Cu.

Dislocation density measurements are reported in Table 3. A RUS-determined dislocation density $\Lambda_{RUS}$ is obtained using Equation (Equation 1), together with a typical dislocation segment length L≈150 nm for Al and L≈230 nm for Cu. The results for the shear wave velocity $v_{T}$ reported above provide a variation between samples of $\Delta \Lambda_{RUS}\approx \left(4-7\right)\times 10^{7}$ mm${}^{-2}$ for the Al group and $\Delta \Lambda_{RUS}\approx \left(3-5\right)\times 10^{7}$ mm${}^{-2}$ for the Cu group. In both cases the associated errors are less than 20%. The XRD-determined dislocation density $\Lambda_{XRD}$, as expected, is lower for annealed samples than for purely rolled ones. However, the associated errors are so large that it is not possible to clearly differentiate between pieces within each group. In any case, the values obtained are of the same order of magnitude of the acoustically obtained values so they do provide a check on the latter method. In Figure 5, we present the quantitative relation between the variations of the nonlinear parameters with respect to the changes in dislocation density. In a first approximation, we obtain that $\Delta \alpha^{\prime }$ and $\Delta \beta^{\prime }$ are linearly dependent of $\Delta \Lambda_{RUS}$. This method then provides a way to obtain dislocation density variations as a function of the changes of the acoustic nonlinear parameters, with a high sensitivity compared to linear measurements. Thus, for a given material and once properly calibrated, one can indeed use the high sensitivity of the nonlinear parameters in order to quantitatively study dislocation proliferation in metals and alloys.

## 4. Discussion

The nonlinear parameter β is defined through $\beta \equiv -[3+\left(C_{111}/C_{11}\right)]$ [25], with $C_{11}$ and $C_{111}$ the second- and third-order longitudinal elastic constants given by $\sigma =C_{11}\epsilon +\left(C_{111}+C_{11}\right)\epsilon^{2}+\cdots$, where σ is stress and ϵ is strain. We already know [14] that *n* dislocation segments of length *L* per unit volume induce a change $\Delta C_{11}$ given by $\Delta C_{11}/C_{11}=-32\Delta \left(nL^{3}\right)/\left(45\pi^{2}\right)$. The influence of dislocations on β has been studied by several authors [37,38,39,40,41]. Since this influence is a small effect, one has that the change induced is proportional to dislocation density: $\Delta C_{111}/C_{111}=B\Delta \left(nL^{3}\right)$, with a dimensionless constant *B* that depends on the geometry and modeling employed. A simple calculation shows
(6)$$\Delta \beta =-\left(\frac{\Delta C_{111}}{C_{111}}-\frac{\Delta C_{11}}{C_{11}}\right)\frac{C_{111}}{C_{11}}\;.$$

Since, for aluminum and copper $C_{111}∼-10C_{11}$ [42], this formula provides, a rationale for understanding the factor of ten higher sensitivity of β to dislocation density, compared to the second order coefficient, as well as its increase, as long as $\Delta C_{111}/C_{111}>\Delta C_{11}/C_{11}$.

The parameter α depends on the coupling between the different normal modes of an elastic sample due to nonlinearities. Chakrapani and Barnard [43] have determined, both theoretically and experimentally, the value of α for a purely longitudinal mode of a thin beam, and have inferred that β=−Kα with K>0. Our measurements of $\alpha^{\prime }$ and $\beta^{\prime }$ are consistent with this result (we remind that from Equations (Equation 3) and (Equation 4), we have $\alpha^{\prime }\propto \alpha$ and $\beta^{\prime }\propto \beta$). In particular, when the dislocation density increases the material is more nonlinear with respect to β, as it increases, but less nonlinear for α as it decreases in its absolute value. However, further investigation and modeling are needed to ascertain a precise formula for the influence of dislocations on the parameter α.

## 5. Conclusions

We have measured the change in the nonlinear parameters $\beta^{\prime }$ and $\alpha^{\prime }$ as a function of the change in dislocation density in copper and aluminum, the change in dislocation density $nL^{3}$ being determined by linear acoustics. We have determined that a change of $nL^{3}$ by a factor of ten leads to a 20–60% change in $\beta^{\prime }$, and to a factor of two to six change in $\alpha^{\prime }$. We also explain the difference in about a factor of ten between the sensitivity of the linear and nonlinear measurements. These results pave the way for the use of nonlinear acoustics as a sensitive, quantitative, probe of dislocation density in metals and alloys.

## Figures and Tables

**Figure 1 materials-11-02217-f001:**
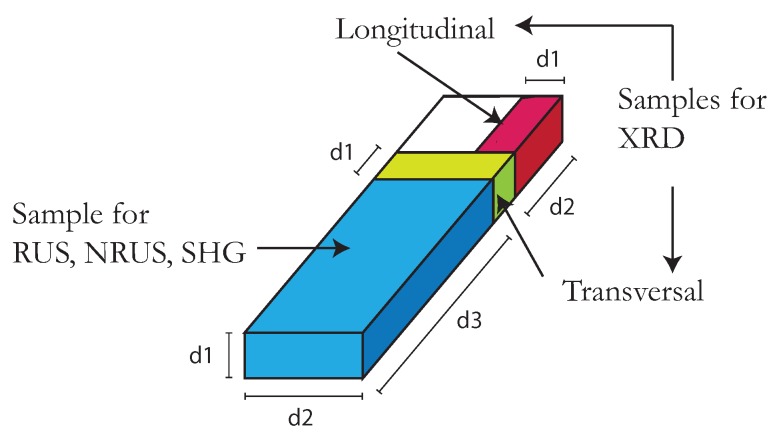
Schematic illustration of samples. For each thermo-mechanical treatment and for both groups, three pieces were cut for the application of acoustic methods and XRD. The longest dimension corresponds to the original’s bar axis and the cold-rolling direction. Consequently, the XRD samples are named longitudinal and transversal.

**Figure 2 materials-11-02217-f002:**
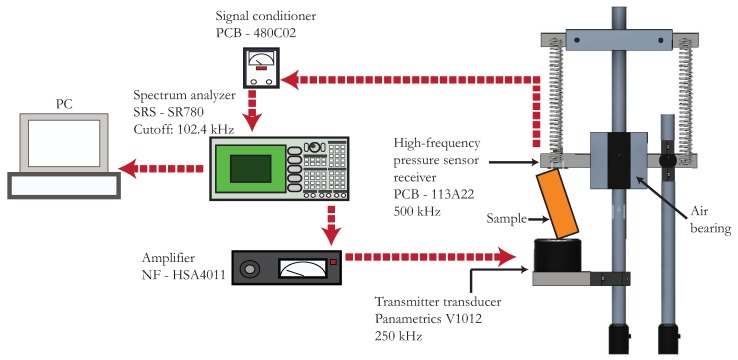
Schematic illustration of the experimental setup used for both RUS and NRUS. The sample is positioned between a contact ultrasonic transducer and a high frequency response pressure sensor. The set of springs and the air bearing ensure that the contact force applied to the sample is very small, which enables to compare the measured resonant frequencies with those of a free-stress parallelepiped. For resonances below 102.4 kHz a spectrum analyzer is used for the frequency sweep. Above this frequency limit, this apparatus is replaced by a National Instruments digital-to-analog acquisition card, model PCI-6251.

**Figure 3 materials-11-02217-f003:**
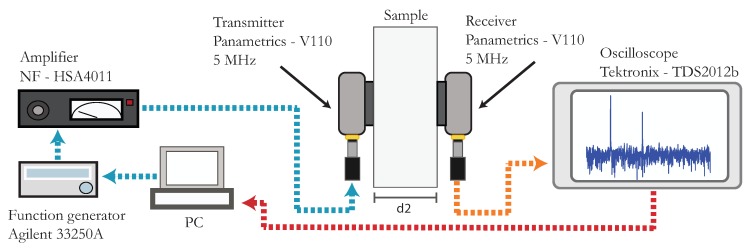
Schematic illustration of the experimental setup used for SHG. A sinusoidal voltage waveform is amplified and used to generate an ultrasonic signal, emited by one of the transducers. The second one receives the transmitted signal and its FFT spectrum is computed by an oscilloscope. Both the fundamental and first harmonic amplitudes are recorded on a personal desktop computer.

**Figure 4 materials-11-02217-f004:**
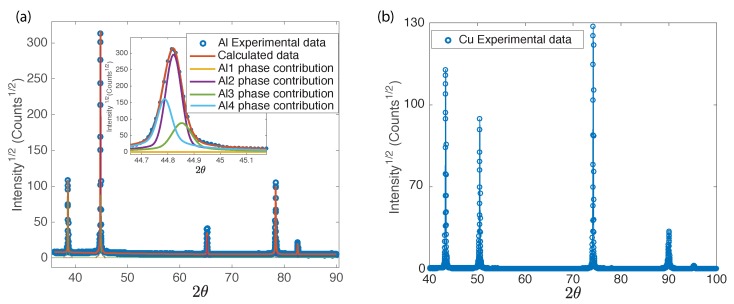
Example of (**a**) Aluminum and (**b**) Copper XRD pattern. (**a**) Five peaks are observed for Al, corresponding to different lattice planes: (111) ($2\theta =38.55^{\circ }$), (200) ($2\theta =44.81^{\circ }$), (220) ($2\theta =65.21^{\circ }$), (311) ($2\theta =78.35^{\circ }$) and (400) ($2\theta =99.22^{\circ }$). Inset: A distribution of crystallite sizes (Al${}_{i},i=1,2,3,4$) contribute to the (200) diffraction peak (shown) as well as to the others (not shown). (**b**) For Cu samples, five peaks are observed in the angular range measured, corresponding to the following lattice planes: (111) ($2\theta =43.37^{\circ }$), (200) ($2\theta =5051^{\circ }$), (220) ($2\theta =74.2^{\circ }$), (311) ($2\theta =90.01^{\circ }$) and (222) ($2\theta =95.23^{\circ }$).

**Figure 5 materials-11-02217-f005:**
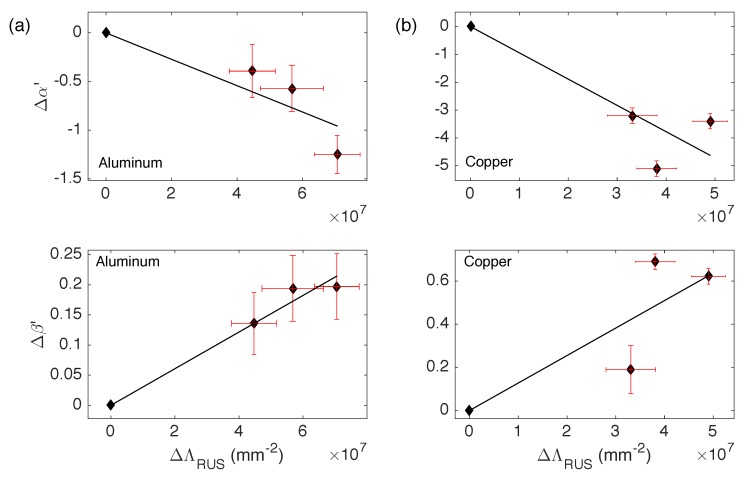
Normalized variations of nonlinear acoustic parameters $\alpha^{\prime }$ and $\beta^{\prime }$ of each sample respect to the purely rolled one as functions of the variations of dislocation density, obtained with the linear measurements. For both groups Al and Cu, $\Delta \Lambda_{RUS}$ are similar, which are obtained from changes in the transverse elastic wave speed $v_{T}$, which are of the order of a few percent. (**a**) For the Al group, $\alpha^{\prime }$ shows changes of 39% to 125% and $\beta^{\prime }$ of 14% to 20%; (**b**) For the Cu group, $\alpha^{\prime }$ shows changes of 320% to 510%, and $\beta^{\prime }$ of 19% to 62%.

**Table 1 materials-11-02217-t001:** Aluminum and Copper group characteristics: rectangular parallelepiped dimensions, mass density and treatments for the four samples of each group. Columns are ordered for decreasing expected dislocation density.

**Aluminum**				
**Parameter**	**Al Roll**	**Al Roll-A15**	**Al Roll-A30**	**Al Roll-A60**
$d_{1}$ (cm)	0.501±0.001	0.503±0.001	0.499±0.001	0.497±0.001
$d_{2}$ (cm)	1.702±0.001	1.704±0.001	1.702±0.001	1.709±0.001
$d_{3}$ (cm)	5.002±0.001	5.004±0.001	5.004±0.001	5.008±0.001
ρ (g/cm${}^{3}$)	2.670±0.006	2.661±0.006	2.672±0.006	2.665±0.006
Treatments				
Rolled	82.8%	82.8%	82.8%	82.8%
Annealed	-	$450{\;}^{\circ }$C ×15 min	$450{\;}^{\circ }$C ×30 min	$450{\;}^{\circ }$C ×60 min
**Copper**				
**Parameter**	**Cu Roll 2**	**Cu Roll-A15**	**Cu Roll-A30**	**Cu Roll-A60**
$d_{1}$ (cm)	0.401±0.001	0.401±0.001	0.401±0.001	0.392±0.001
$d_{2}$ (cm)	1.700±0.001	1.700±0.001	1.702±0.001	1.701±0.001
$d_{3}$ (cm)	5.006±0.001	4.999±0.001	5.000±0.001	4.999±0.001
ρ (g/cm${}^{3}$)	8.882±0.023	8.883±0.023	8.901±0.023	8.898±0.023
Treatments				
Rolled	88.3%	88.3%	88.3%	88.3%
Annealed	-	$850{\;}^{\circ }$C ×15 min	$850{\;}^{\circ }$C ×30 min	$850{\;}^{\circ }$C ×60 min

**Table 2 materials-11-02217-t002:** Acoustic parameters, both linear and nonlinear, obtained for each group of samples compared and contrasted. Nonlinear parameters $\alpha^{\prime }$ and $\beta^{\prime }$ exhibit a considerably higher change from sample to sample than the linear parameter $v_{T}$. Errors are obtained by standard deviation of ten measurements with each method. See text for symbol definition.

**Aluminum**			
**Treatment**	$v_{T}$ **(m/s)**	$\frac{\alpha^{\prime }}{10^{-4}}$ **(V** ${}^{-1}$ **)**	$\beta^{\prime }$ **(V** ${}^{-1}$ **)**
Roll A60	3116±4	−39±8	0.42±0.02
Roll A30	3130±7	−44±7	0.39±0.02
Roll A15	3146±4	−63±5	0.39±0.02
Roll	3065±4	−28±5	0.49±0.01
**Copper**			
**Treatment**	$v_{T}$ **(m/s)**	$\frac{\alpha^{\prime }}{10^{-4}}$ **(V** ${}^{-1}$ **)**	$\beta^{\prime }$ **(V** ${}^{-1}$ **)**
Roll A60	2294±6	−168±21	0.90±0.10
Roll A30	2304±4	−244±31	0.35±0.01
Roll A15	2326±3	−176±18	0.42±0.01
Roll	2229±4	−40±10	1.11±0.03

**Table 3 materials-11-02217-t003:** Comparison of XRD and RUS measurements of relative dislocation density for the Al and Cu samples. Errors for XRD measurements are calculated with the contribution of the Rietveld refinement results and the statistical error from the repetition of the experiment in two pieces of the same sample (longitudinal and transversal). These errors are large and preclude a sample-to-sample comparison. By contrast, the errors associated with the acoustic measurements are sufficiently small that a quantitative comparison can be confidently provided.

**Aluminum**		
**Compared Samples**	$\frac{\Delta \Lambda_{\mathrm{XRD}}}{10^{7}}$ **(mm** ${}^{-2}$ **)**	$\frac{\Delta \Lambda_{\mathrm{RUS}}}{10^{7}}$ **(mm** ${}^{-2}$ **)**
Roll & Roll A60	1.24±1.47	4.47±0.70
Roll & Roll A30	0.87±1.35	5.68±0.96
Roll & Roll A15	0.42±7.12	7.07±0.69
**Copper**		
**Compared Samples**	$\frac{\Delta \Lambda_{\mathrm{XRD}}}{10^{7}}$ **(mm** ${}^{-2}$ **)**	$\frac{\Delta \Lambda_{\mathrm{RUS}}}{10^{7}}$ **(mm** ${}^{-2}$ **)**
Roll & Roll A60	2.34±21.74	3.31±0.51
Roll & Roll A30	4.73±19.35	3.81±0.41
Roll & Roll A15	5.04±19.0	4.90±0.35

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
