# Peer review of "Linear Versus Nonlinear Acoustic Probing of Plasticity in Metals: A Quantitative Assessment"

_materials, 2018, doi:10.3390/ma11112217_

Reviewer 1 Report

An interesting manuscript on the dislocation density determination by using resonant ultrasound spectroscopy (RUS, NRUS) and SHG methods. Some correction can improve a science value of the manuscript. The considered points are listed below:

1)Equipments used for testing? Specify.

2) Dimensions of the samples? Add a schematic illustration of the cold-rolled (CR) and CR+annealed specimens.

3) P3L68....Then, three samples of each group were annealed at about 70% of their melting point.....-Specify melting point (temperature) of investigated materials.

4) Graphical comparison of the abtained data for both materials. Maybe new chapter: Discussion....?

5) Why is the Figure 2 located after Authors Contributions and Acknowledgemet?

Author Response

Response to referee 1

An interesting manuscript on the dislocation density determination by using resonant ultrasound spectroscopy (RUS, NRUS) and SHG methods. Some correction can improve a science value of the manuscript. The considered points are listed below:

Answer: we thank the referee for his/her positive feedback.

1)Equipments used for testing? Specify.

Answer: we have added two new figures (figure 2 and 3) providing details on the experimental setups, procedures, and equipement (brands and models).

2) Dimensions of the samples? Add a schematic illustration of the cold-rolled (CR) and CR+annealed specimens.

Answer: We have added a new figure (figure 1) with a schematic representation of the sample cuts used for the acoustic and XRD measurements. We have also added a new table (table 1) with details of each sample for both Al and Cu groups (dimensions, mass density and thermo-mechanical treatments).

3) P3L68....Then, three samples of each group were annealed at about 70% of their melting point.....-Specify melting point (temperature) of investigated materials.

Answer: We have specified in the text and in the new table 1 the precise annealing temperatures, as well as the melting points in the text.

4) Graphical comparison of the abtained data for both materials. Maybe new chapter: Discussion....?

Answer: concerning the presentation of results and their discussion, we have now separated the previous section in two sections (Results and a new one Discussion).

5) Why is the Figure 2 located after Authors Contributions and Acknowledgemet?

Answer: This is just an error of the latex compilation. It is now placed in a better place (the new figure 5).

Reviewer 2 Report

The paper presents a convincing analysis of the application of nonlinear acoustics to survey the variation of the dislocation density in crystals. In my opinion, it may be published as is. I only have a couple of minor suggestions to improve readability of the paper for a wider audience:

1. Whereas many notions used in acoustics (e.g., Rayleigh waves) are mentioned in general courses of physics, plasticity of crystals is a more specialized branch. It would be useful to specify in Line 31 that the Taylor’s rule relates the flow stress to the dislocation density.

2. I will be clearer if Ref. [33] appears after mentioning the March-Dollase model in line 92 and not at the end of the paragraph.

By the way, there is no need to insist on “unmistakable” difference in lines 51 and 53. Il line 53, it would be enough to write “this difference” instead of “this unmistakable difference”.

Author Response

Response to referee 2

Comments and Suggestions for Authors

The paper presents a convincing analysis of the application of nonlinear acoustics to survey the variation of the dislocation density in crystals. In my opinion, it may be published as is. I only have a couple of minor suggestions to improve readability of the paper for a wider audience:

Answer: we thank the referee for his/her positive feedback.

1. Whereas many notions used in acoustics (e.g., Rayleigh waves) are mentioned in general courses of physics, plasticity of crystals is a more specialized branch. It would be useful to specify in Line 31 that the Taylor’s rule relates the flow stress to the dislocation density.

Answer: a phrase has been added (in line 32 now).

2. I will be clearer if Ref. [33] appears after mentioning the March-Dollase model in line 92 and not at the end of the paragraph.

Answer: the reference has been moved as suggested.

By the way, there is no need to insist on “unmistakable” difference in lines 51 and 53. Il line 53, it would be enough to write “this difference” instead of “this unmistakable difference”.

Answer: We agree, the phrase has been changed accordingly.